# Analysis of the Alignment of Non-Random Patterns of Spin Directions in Populations of Spiral Galaxies

Lior Shamir

Kansas State University, Manhattan, KS 66506, USA; lshamir@mtu.edu

**Abstract:** Observations of non-random distribution of galaxies with opposite spin directions have recently attracted considerable attention. Here, a method for identifying cosine-dependence in a dataset of galaxies annotated by their spin directions is described in the light of different aspects that can impact the statistical analysis of the data. These aspects include the presence of duplicate objects in a dataset, errors in the galaxy annotation process, and non-random distribution of the asymmetry that does not necessarily form a dipole or quadrupole axes. The results show that duplicate objects in the dataset can artificially increase the likelihood of cosine dependence detected in the data, but a very high number of duplicate objects is required to lead to a false detection of an axis. Inaccuracy in galaxy annotations has relatively minor impact on the identification of cosine dependence when the error is randomly distributed between clockwise and counterclockwise galaxies. However, when the error is not random, even a small bias of 1% leads to a statistically significant cosine dependence that peaks at the celestial pole. Experiments with artificial datasets in which the distribution was not random showed strong cosine dependence even when the data did not form a full dipole axis alignment. The analysis when using the unmodified data shows asymmetry profile similar to the profile shown in multiple previous studies using several different telescopes.

**Keywords:** cosmology; galaxies; large-scale structure

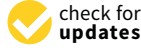



## 1. Introduction

The contention that the spin directions of spiral galaxies are distributed in a non-random manner [1–7] has been considered a certain mystery in astronomy in the past decade. While early experiments with manually annotated galaxies were limited by the size of the data that can be processed [1,8], the availability of autonomous digital sky surveys combined with computational methods that can process a very high number of galaxies provided far larger datasets, providing certain evidence that the spin directions of spiral galaxies is not necessarily randomly distributed [2,5,6]. Analysis of these large automatically annotated datasets provided certain evidence that the asymmetry is statistically significant, and changes based on the direction of observation, and the redshift [5,6]. That has been shown with data from different digital sky surveys such as the Panoramic Survey Telescope and Rapid Response System (Pan-STARRS) and Sloan Digital Sky Survey (SDSS), both providing a similar profile of asymmetry [6].

These experiments also included many tests for possible errors, such as a possible dependence between the accuracy of the annotation and the size of the objects or their redshift, presence of duplicate objects in the training set, and "sanity checks" such as repeating the experiments after mirroring the galaxies images. These experiments did not involve elements of human annotation or machine learning to avoid the effect of human perceptual bias or complex non-intuitive data-driven rules often used by machine learning systems.

Smaller-scale experiments with manually annotated galaxies also showed unexplained correlations between the spin directions of galaxies [9], even when the galaxies are too far to have gravitational interactions [7]. These observations might conflict with the standard

cosmology when assuming Newtonian gravity. However, assuming modifications of the Newtonian dynamics (MOND) models can explain longer span of the gravitational impact, while also providing an explanation to the Keenan–Barger–Cowie (KBC) void in the context of the standard model [10].

It has been shown that the spin directions of galaxies correlate with the alignment of the cosmic filament they are part of [11,12]. It should be noted that the spin is not the same as the two-headed alignment, for which the spins show no preference between parallel and antiparallel to the direction of the alignment. Several studies of dark matter simulation also showed a correlation between the spin and the large-scale structure [13–15], and the strength of the correlation is linked to the color and stellar mass of the galaxies [16]. These links were associated with halo formation [17], proposing that the spin in halo progenitors is aligned with the large-scale structure in the early universe [18]. While it might seem intuitive to assume that the spin direction of a galaxy is aligned with its host halo, it has been suggested that a galaxy can also spin in a different direction compared to its halo [16].

The recent consistent evidence related to the alignment of galaxy spin directions in the context of the large-scale structure reinforce the need to study the distribution of the spin directions of spiral galaxies beyond the null-hypothesis of a fully random distribution. Such examination should also consider the possibility that the distribution of the spin directions of spiral galaxies is related to the large-scale structure, possibly in the form of cosmological-scale axes. In this paper, a method for identifying cosine dependence that forms a dipole or quadrupole axes in a dataset of galaxies annotated by their spin directions is discussed. Several aspects that can affect the analysis are tested, including the presence of duplicate objects in the dataset, inaccurate annotations of the spin directions of the galaxies, and non-random distribution of the spin directions that does not fit a full dipole alignment. The analysis is demonstrated using a dataset of $\sim 7.7 \times 10^4$ SDSS galaxies.

## 2. Analysis Method

Assuming that the distribution of spin directions of spiral galaxies exhibits a cosmological-scale dipole axis, it is expected that the spin direction distribution would have a non-random cosine dependence. In other words, statistically significant fitness of the spin directions of the spiral galaxies into cosine dependence can be an indication of dipole alignment of the spin directions.

The cosine dependence between the angle and the spin directions of the galaxies can be computed from each $(\alpha, \delta)$ coordinates in the sky. For each possible integer $(\alpha, \delta)$ combination, the angular distance between $(\alpha, \delta)$ to all galaxies in the dataset is computed. Then, $\chi^2$ statistics can be used to fit the spin direction distribution to cosine dependence. That is done by fitting $d \cdot |\cos(\phi)|$ to $\cos(\phi)$, where $\phi$ is the angular distance between the galaxy and $(\alpha, \delta)$, and $d$ is a number within the set $\{-1, 1\}$, such that $d$ is 1 if the galaxy spins clockwise, and $-1$ if the galaxy spins counterclockwise.

The $\chi^2$ computed when the $d$ of each galaxy is assigned the actual spin directions of the galaxies is compared to the average of the $\chi^2$ when computed in $10^3$ runs such that the $d$ of each galaxy is assigned with a random number within $\{-1,1\}$. After repeating $10^3$ runs, each with a different set of random values, the mean and standard deviation of the $\chi^2$ of all runs can be determined. Then, the $\sigma$ difference between the $\chi^2$ computed with the actual spin directions and the mean $\chi^2$ computed with the random spin directions provide the probability to have a dipole axis in that $(\alpha, \delta)$ combination. Computing the probability for each possible integer $(\alpha, \delta)$ in the sky can provide an all-sky analysis of the probability of a dipole axis. The same method can also be used to fit the distribution to quadrupole alignment, by fitting the distribution of the spin directions to $\cos(2\phi)$.

## 3. Data

The dataset used in this study is based on the dataset of photometric objects imaged by Sloan Digital Sky Survey, from which duplicate photometric objects were removed.

The purpose of that dataset was to examine evidence of photometric differences between galaxies with opposite spin directions observed in a much smaller dataset [4]. The dataset contains 740,908 relatively bright ($g < 19$) and large (Petrosian radius $> 5.5''$) objects identified as galaxies by the SDSS photometric pipeline. Setting the radius and magnitude limit was necessary due to the very large number of photometric objects in SDSS, making it impractical to download and analyze all SDSS photometric objects.

The galaxies were annotated by their spin direction by applying the Ganalyzer algorithm [19,20]. In summary, Ganalyzer works by first converting the galaxy image into its radial intensity plot, then identifies the arms of the galaxy by detecting the peaks in the radial intensity plot, and then applies a linear regression to the peaks to identify the slope, which directly reflect the spin direction of the arm. Unlike numerous recent galaxy annotation methods, Ganalyzer's main advantage is that it is not based on machine learning or deep learning algorithms that rely on complex non-intuitive data-driven rules. As will be discussed later in this paper, certain noise in the data does not have a major impact on the detection. However, even a small but consistent bias in the annotation of the galaxies can lead to a statistically significant non-random alignment of the galaxies. Since such subtle biases are often difficult to identify, the analysis method needs to be a symmetric method. More information about Ganalyzer is available in [19], and the process of the galaxy annotation is described in [21,22].

Since not all galaxies have identifiable spin directions, Ganalyzer does not assign a spin direction to all galaxies, and annotates some galaxies as "undetermined". From the dataset of 740,908, 172,883 objects were assigned a spin direction. For the removal of photometric objects that are part of the same galaxies, photometric objects that were closer than $0.01°$ to another object in the dataset were removed. Removing all duplicate objects provided a dataset of 77,840 galaxies, available at http://people.cs.ksu.edu/~lshamir/data/assymdup. The distribution of the exponential r magnitude is shown in Figure 1, and Figure 2 shows the distribution of the redshift of the galaxies that have spectra in SDSS.

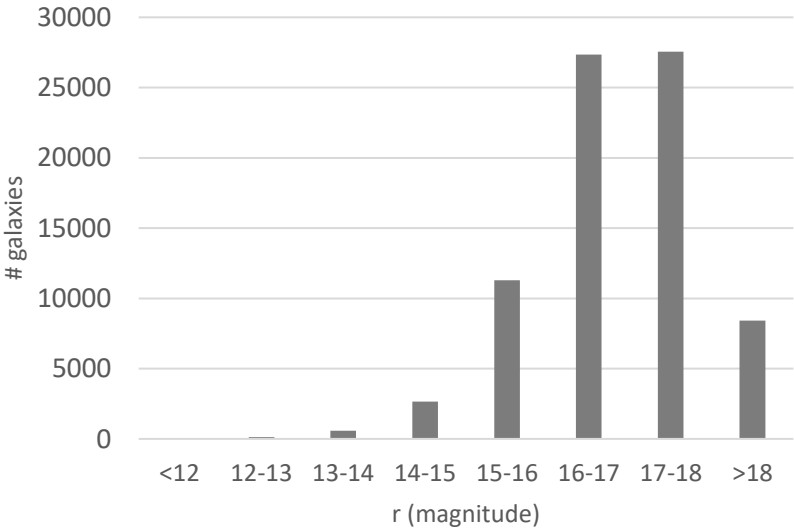

**Figure 1.** The distribution of the exponential r magnitude of the galaxies in the dataset.

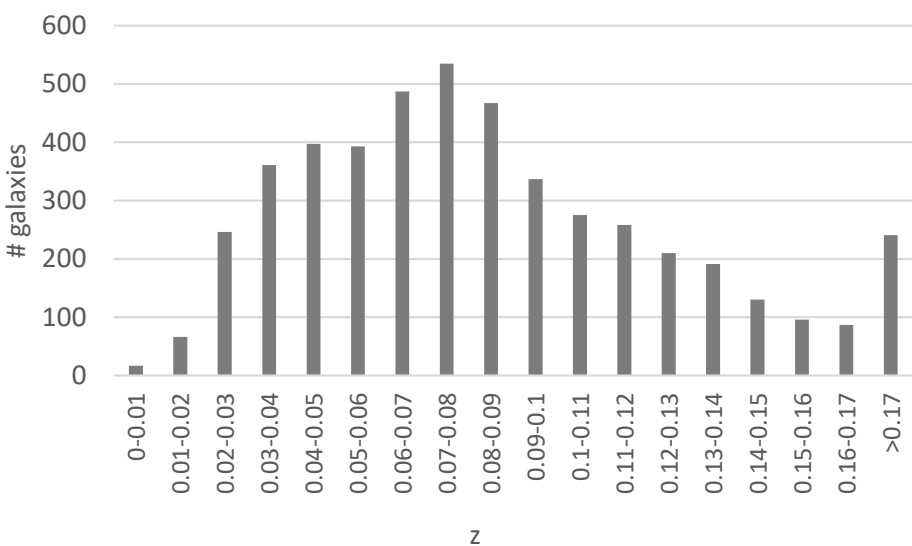

**Figure 2.** The distribution of the redshift of the galaxies in the dataset that have spectra.

Of the 77,840 galaxies in the dataset, 39,187 galaxies spin clockwise, and 38,653 had a counterclockwise spin, showing a ~1.4% difference. Using binomial distribution, the probability to have ~1.4% more clockwise galaxies compared to counterclockwise galaxies by chance is 0.0275, and the two-tailed probability is 0.055.

As was shown in [2], it is possible that the difference between the number of clockwise and counterclockwise galaxies changes based on the direction of observation.

Figure 3 shows the asymmetry $A$ between the number of clockwise galaxies and the number of counterclockwise galaxies in different RA ranges. The asymmetry $A$ is defined by $A = \frac{N_{cw} - N_{ccw}}{N_{cw} + N_{ccw}}$. Each bar shows the asymmetry between the number of clockwise galaxies and counterclockwise galaxies in a certain RA range. The declination is not used in this visualization, but the galaxies are separated by their RA only. The galaxies are not separated by their declination ranges, and therefore each RA slice is averaged by the declination. The figure shows differences in the number of clockwise and counterclockwise galaxies in different RA ranges. When separating the dataset into two RA hemispheres, the strongest asymmetry is observed in the hemisphere centered at ($\alpha = 160°$). In that hemisphere there are 24,648 galaxies with clockwise spin and just 23,958 galaxies with counterclockwise spin. The one-tailed probability of having such a difference or greater by chance is 0.00086, and the two-tailed probability is 0.0017. The opposite hemisphere has a higher number of galaxies with counterclockwise spin direction. Although that asymmetry is not significant, it also does not conflict with the asymmetry in the other hemisphere for the assumption that these two hemisphere exhibits a dipole. The weaker signal in one hemisphere can be because most of the SDSS galaxies are in the hemisphere toward right ascension 180°. The uneven distribution of the SDSS galaxies in the sky can also lead to a bias of the real asymmetry compared to the asymmetry shown using SDSS data.

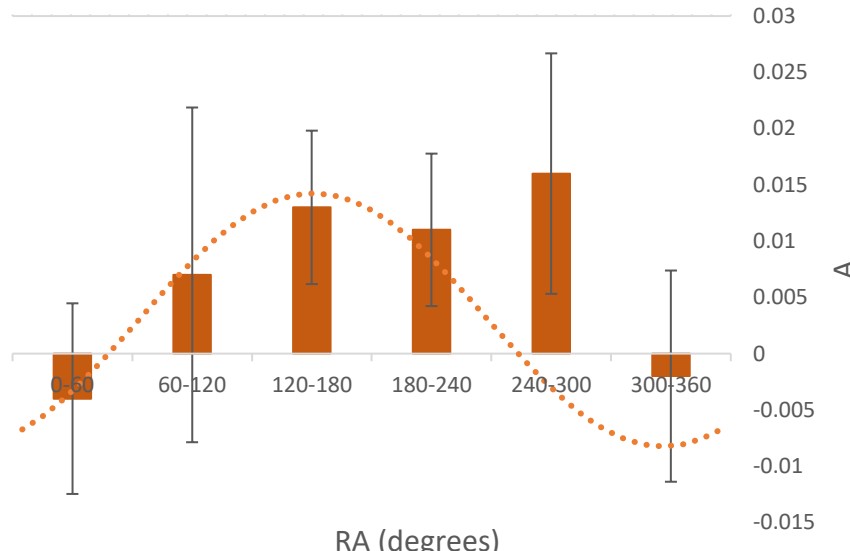

**Figure 3.** The asymmetry $A$ between the number of galaxies that spin clockwise and the number of galaxies that spin counterclockwise in different 60° RA ranges. The asymmetry $A$ is defined by $A = \frac{N_{cw} - N_{ccw}}{N_{cw} + N_{ccw}}$. Each bar shows the asymmetry between clockwise and counterclockwise galaxies in a different RA range. The error bars are the normal distribution standard error of $\frac{1}{\sqrt{N}}$, where $N$ is the total number of galaxies in the RA range. The dashed line shows the cosine of the RA centered at RA = 160°.

Applying the analysis to the data described in Section 2 provided a dipole axis with maximum statistical strength of $2.56\sigma$. The most likely location of the dipole axis was identified as described in Section 2 at ($\alpha = 165°$, $\delta = 40°$). The $1\sigma$ error range is ($90°$, $240°$) for the right ascension, and ($-35°$, $90°$) for the declination. Interestingly, the most likely dipole axis is not particularly far from the location of the most likely dipole reported in [2] at ($\alpha = 132°$, $\delta = 32°$), and well within the $1\sigma$ error range.

The dataset used in [2] also contained galaxies with similar radius and magnitude limits as the dataset used here. Figure 4 shows the likelihood of the dipole axis from each possible integer ($\alpha$, $\delta$) combination in increments of five degrees. Figure 5 shows the same analysis, when the Mollweide projection is centered around ($\alpha = 165°$, $\delta = 40°$).

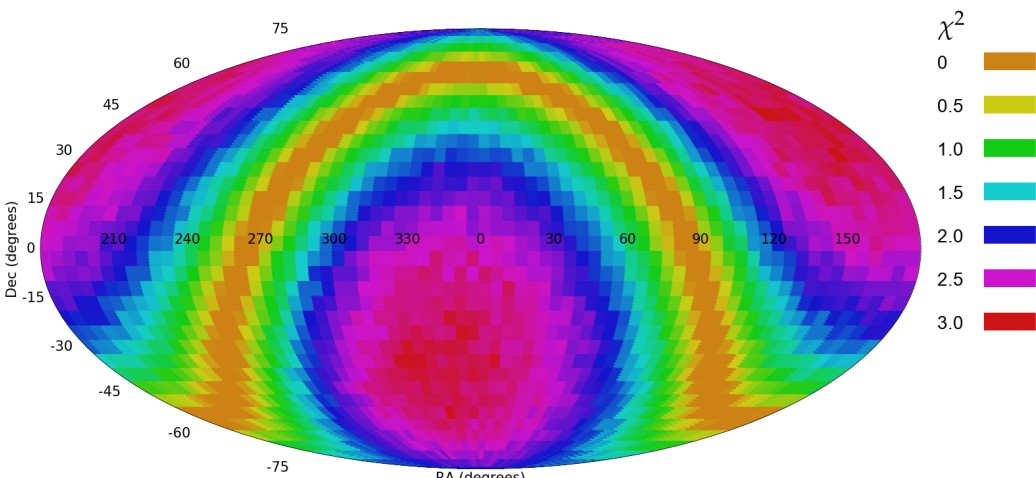

**Figure 4.** The $\chi^2$ probability of a dipole axis in spin directions of the galaxies from different ($\alpha$, $\delta$) combinations.

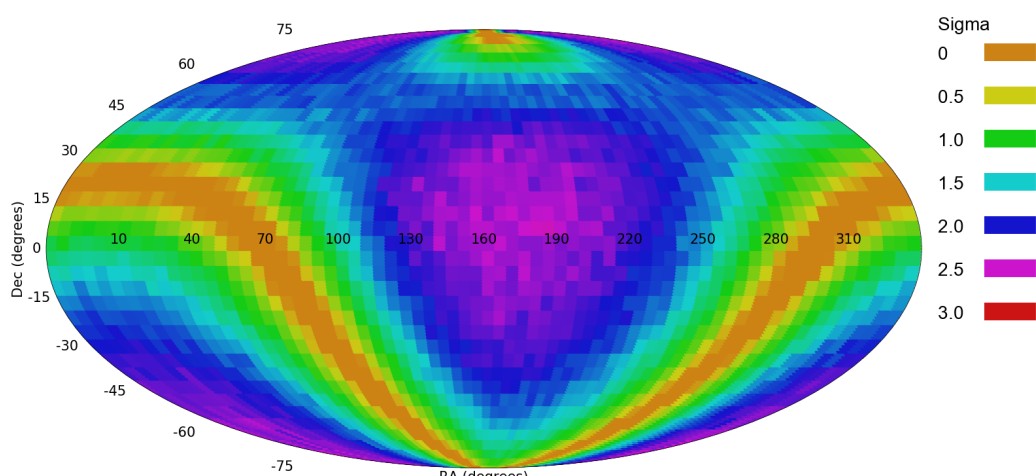

**Figure 5.** The $\chi^2$ probability of a dipole axis in spin directions of the galaxies from different $(\alpha, \delta)$ combinations, centered at $(\alpha = 165°, \delta = 40°)$.

Figure 6 shows the probability of a quadrupole axis from all possible integer $(\alpha, \delta)$ combinations. The most likely axis is identified at $(\alpha = 355°, \delta = 45°)$, with $1\sigma$ error range of $(305°, 60°)$ for the RA, and $(15°, 75°)$ for the declination.

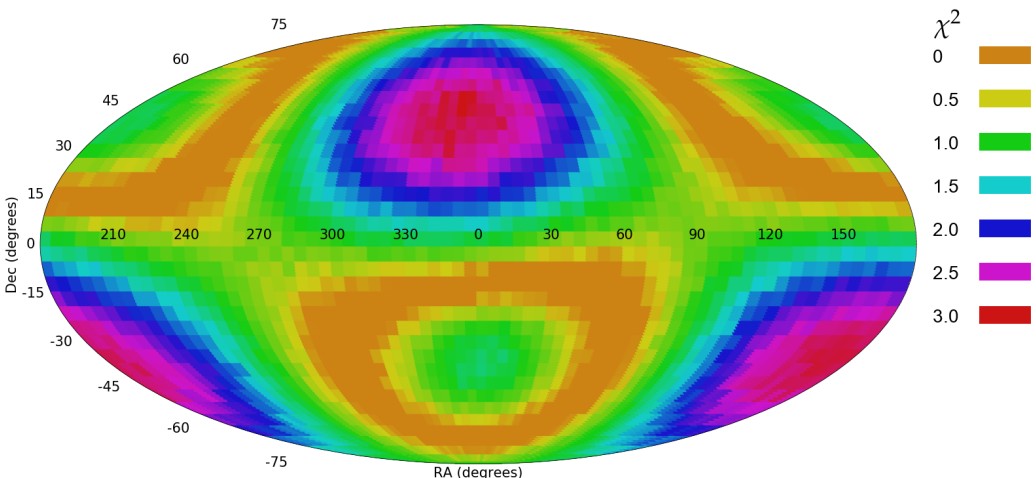

**Figure 6.** The probability of a quatdrupole axis from different $(\alpha, \delta)$.

When the galaxies are assigned with random spin directions, the asymmetry becomes statistically insignificant. Figure 7 shows the differences in different RA ranges when using the same dataset, but when the spin directions of the galaxies are random. Figure 8 shows the likelihood of the dipole axis when the galaxies are assigned with random spin directions. The probability of the most likely axis dropped to $0.78\sigma$.

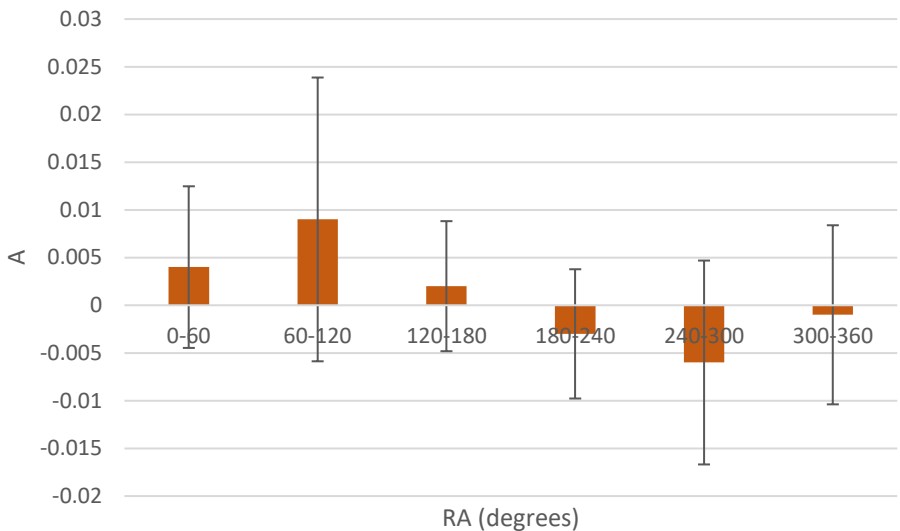

**Figure 7.** The asymmetry between the number of galaxies that spin clockwise and the number of galaxies that spin counterclockwise in different RA ranges when the galaxies are assigned with random spin directions.

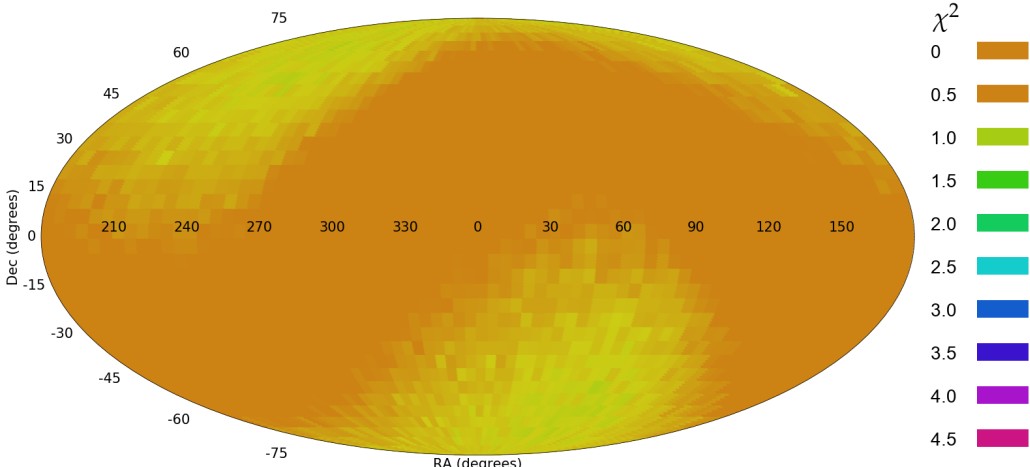

**Figure 8.** The $\chi^2$ probability of the dipole axis when the galaxies are assigned with random spin directions.

The dataset can be separated to galaxies with g magnitude of less than 18, and *g* magnitude greater than 18. That separation provides two orthogonal datasets of galaxies. The number of galaxies with exponential *g* magnitude of less than 18 is 46,052, while 31,798 galaxies had exponential g magnitude greater than 18. Figures 9 and 10 show the analysis for galaxies with g magnitude lower than 18, and g magnitude greater than 18, respectively.

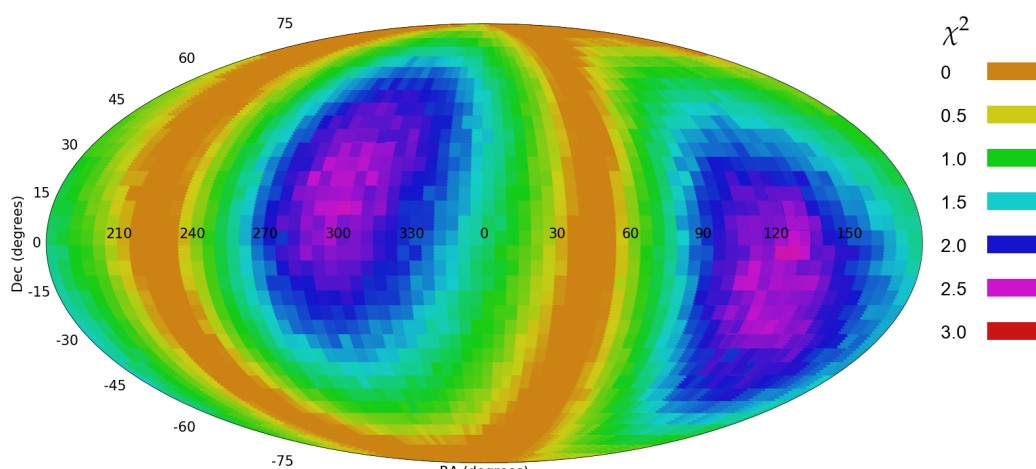

**Figure 9.** The $\chi^2$ probability of the dipole axis when the dataset in limited to galaxies with exponential g magnitude of less than 18.

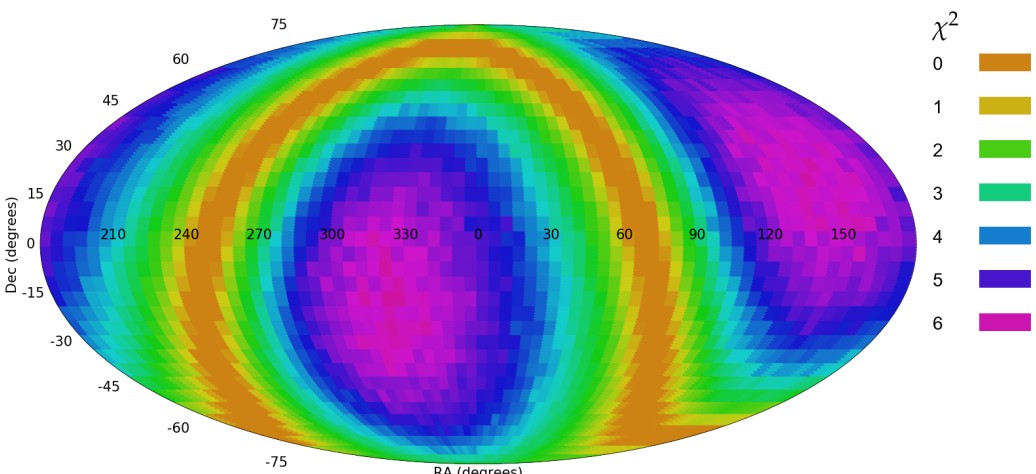

**Figure 10.** The $\chi^2$ probability of the dipole axis when the dataset in limited to galaxies with exponential g magnitude greater than 18.

As the figures show, although the two datasets are completely orthogonal, they provide fairly similar profiles. The most likely axis when the galaxies are limited to $g < 18$ is at ($\alpha = 130°$, $\delta = -10°$), with probability of $2.43\sigma$. When the galaxies are limited to $g > 18$, the most likely axis is at ($\alpha = 150°$, $\delta = 25°$), with probability of $5.57\sigma$. That shows that while the two orthogonal subsets show fairly similar locations of the most likely dipole axis, the statistical signal is stronger when the galaxies have higher g magnitude. Since the magnitude is correlated with the redshift, that agrees with the previous observation that the asymmetry grows when the redshift gets higher [5,6].

## 4. Impact of Duplicate Objects in the Dataset

The dataset described in Section 3, as well as the datasets used in [2,5,6], did not contain duplicate objects. However, when working with photometric measurements of extended objects, a single galaxy can have more than one photometric object in the dataset. To test the impact of duplicate objects several experiments were made by artificially adding duplicate objects to the dataset. In the first experiment, each galaxy in the dataset was duplicated. Naturally, the right ascension, declination, and spin direction of the duplicated galaxy matched the spin direction of the original galaxy. Figure 11 shows the asymmetry between the number of clockwise and counterclockwise galaxies in different RA ranges.

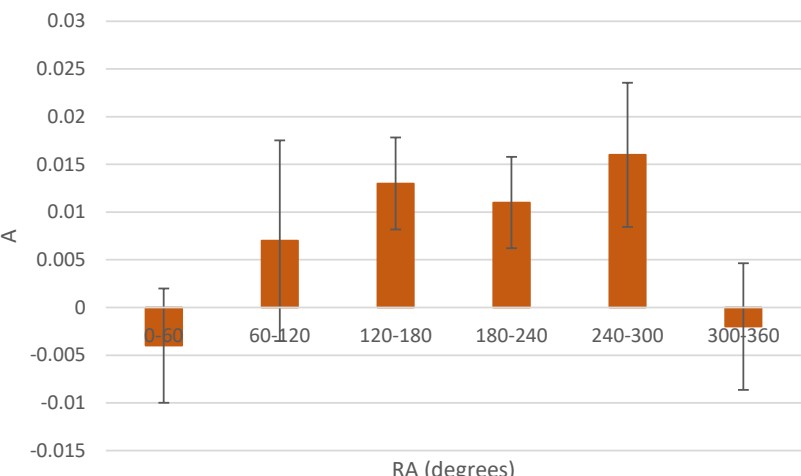

**Figure 11.** The asymmetry between the number of clockwise and counterclockwise galaxies in different RA ranges such that each galaxy in the dataset is duplicated.

As expected, the asymmetry in Figure 11 is identical to the asymmetry in Figure 3. However, the difference is in the error bars, as the standard error is smaller in the dataset that contains duplicate objects. With one duplicate object for each galaxy, the probability to have more clockwise galaxies in the hemisphere around ($\alpha = 160°$) is $<10^{-5}$. That shows that adding duplicate objects does not change the distribution of the galaxies or the profile of the asymmetry, but it increases the statistical significance by increasing the number of galaxies. Figure 12 shows the statistical strength of a dipole axis in different ($\alpha, \delta$). The most likely axis is identified at the same location as in Figure 4, but the statistical signal of the dipole axis increased to $3.62\sigma$.

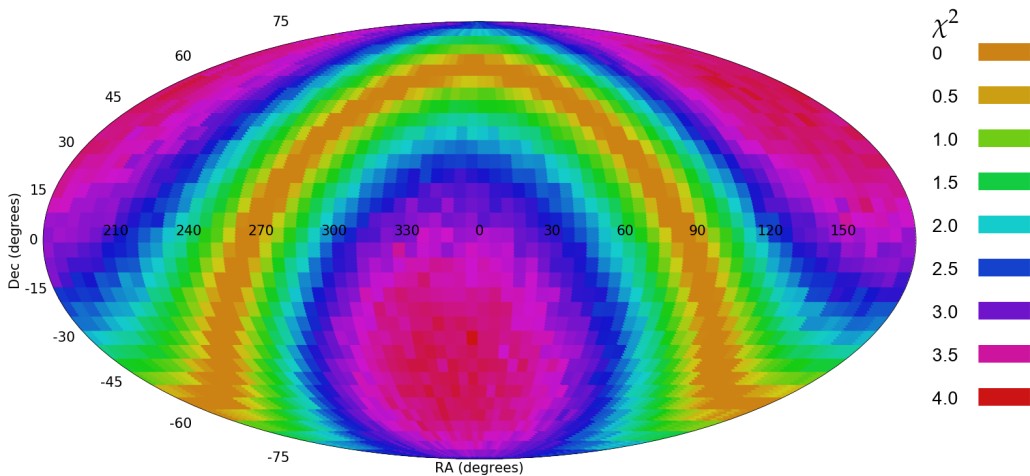

**Figure 12.** The $\chi^2$ probability of a dipole axis in spin directions when each galaxy in the dataset is duplicated.

The graph shows that artificial duplicate objects can strengthen the statistical signal when the spin directions of the galaxies in the original dataset form a statistically significant dipole, as shown in Section 3. To test the impact of duplicate objects in a dataset that does not have signal of parity violation between galaxies with opposite spin directions, an experiment was done with a dataset of galaxies assigned with random spin directions, but each galaxy was duplicated, providing a dataset twice as large as the original dataset.

Figure 13 shows the statistical significance of a dipole axis at different ($\alpha, \delta$) combinations. The maximum dipole axis has statistical strength of $1.93\sigma$. That shows that duplicates in the dataset can lead to statistically significant asymmetry in a dataset, even if the original

dataset has no asymmetry between the number of clockwise and counterclockwise galaxies. However, the number of duplicates needs to be very large, and the dataset needs to be far larger than the original dataset to have an asymmetry that becomes statistically significant.

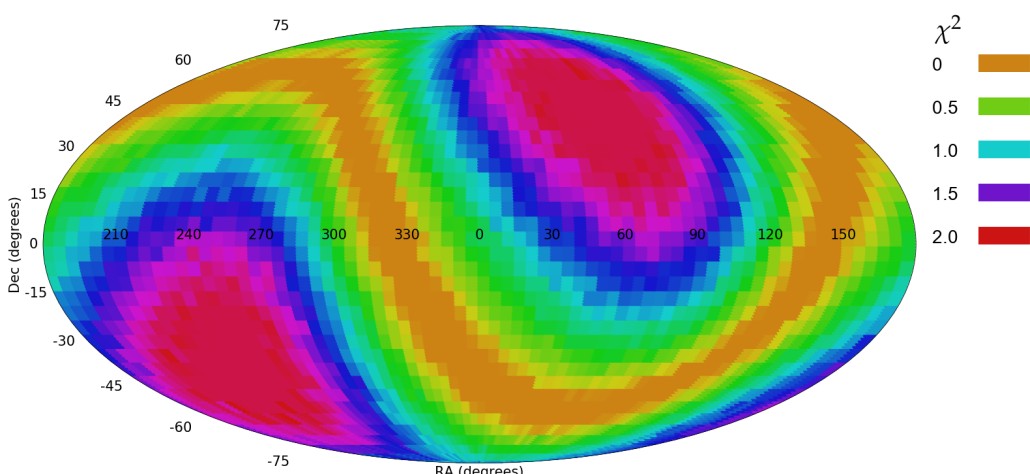

**Figure 13.** The $\chi^2$ probability of a dipole axis in spin directions when the spin directions are random and each galaxy in the dataset is duplicated five times.

## 5. Error in the Galaxy Annotation

Spiral galaxies have complex morphology, and the identification of the spin direction of a galaxy is therefore not a straightforward task. It has been shown that manual identification is heavily biased by the human perception, and therefore datasets annotated manually can be systematically biased [8,23]. Despite the recent advancements in the automatic annotation of galaxy images, galaxy annotation algorithms still do not provide perfect accuracy. The dataset described in Section 3 was created by rejecting galaxy images that were not classified with high certainty, leading to the sacrifice of the majority of the spiral galaxies of the initial dataset.

Since automatic image analysis is a complex task, it is possible that image analysis algorithms have a certain error rate in their classification. Such error rate can also be affected by the size and brightness of the objects, as small and faint objects tend to be more difficult to analyze by both machines and humans. If the galaxy annotation algorithm has a certain rate of misclassified galaxies, the asymmetry $A$ in a certain part of the sky can be defined by $A = \frac{(N_{cw}+E_{cw})-(N_{ccw}+E_{ccw})}{N_{cw}+E_{cw}+N_{ccw}+E_{ccw}}$, where $E_{cw}$ is the number of counterclockwise galaxies classified incorrectly as clockwise, and $E_{ccw}$ is the number of clockwise galaxies classified incorrectly as counterclockwise. If the galaxy classification algorithm is symmetric, the number of counterclockwise galaxies misclassified as clockwise is expected to be roughly the same as the number of clockwise galaxies missclassified as counterclockwise. Assuming $E_{cw} = E_{ccw}$, the asymmetry can be defined as $A = \frac{N_{cw}-N_{ccw}}{N_{cw}+E_{cw}+N_{ccw}+E_{ccw}}$. Since $E_{cw}$ and $E_{ccw}$ cannot be negative, a higher rate of misclassified galaxies is expected to make the asymmetry $A$ lower. Therefore, misclassified galaxies are not expected to exhibit themselves in the form of asymmetry, as long as the classification algorithm is symmetric.

To test the impact of misclassified galaxies empirically, several experiments were performed to test the impact of galaxies that are annotated incorrectly. In the first experiment, 25% of the galaxies were randomly selected, and each was assigned a random spin direction, meaning that $\sim$12.5% of the galaxies were assigned with an incorrect spin direction. The two-tailed chance of a difference between galaxies with clockwise and counterclockwise spin directions is 0.052, and the maximum dipole axis has statistical signal of 1.64$\sigma$. As expected, when 50% of the galaxies are assigned with random spin directions the statistical significance of the dipole axis drops to 1.44$\sigma$.

That shows that inaccuracy in the annotation of the galaxies results in weaker statistical signal. However, the experiment was performed such that the incorrect annotations were

distributed randomly between clockwise and counterclockwise galaxies. To test the impact of systematic bias in the annotations, randomly selected 2% of the galaxies were assigned with clockwise spin direction regardless of their actual spin direction. That means that ~1% of the counterclockwise galaxies were assigned with clockwise spin direction. Figure 14 shows the statistical signal of a dipole axis at different $(\alpha, \delta)$ coordinates. The most likely dipole axis was identified at $\delta = 90°$, with $4.42\sigma$.

Figure 15 shows the same graph created from the same dataset such that 2% of the counterclockwise galaxies were assigned with clockwise spin directions. The statistical significance of the dipole axis in that case elevates to $9.21\sigma$, and the strongest axis was detected at $(\delta = 90°)$. These graphs show that while random incorrect annotations have relatively small impact and lead to weaker signal, even a small rate of consistently incorrect annotations leads to strong statistical signal of a dipole axis. Fitting the spin directions to a quadrupole alignment provides statistical signal of $7.12\sigma$ as shown in Figure 16, but does not identify two strong axes.

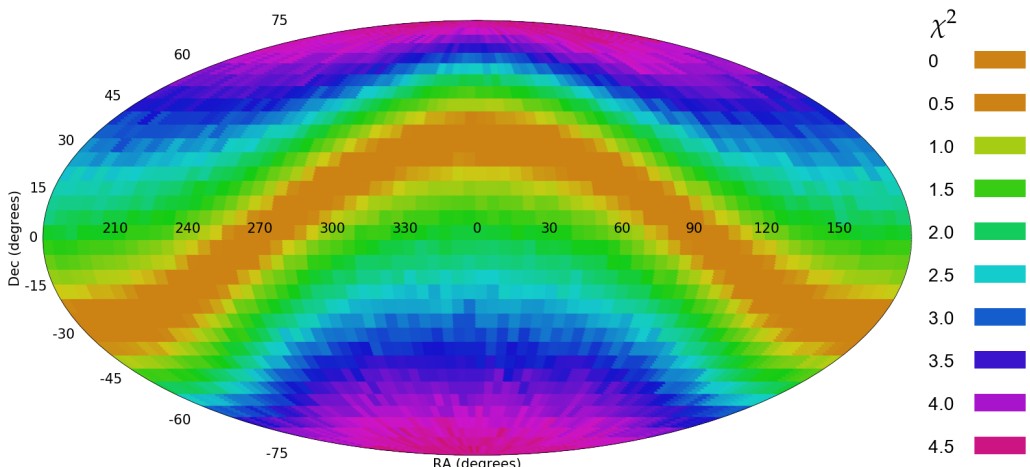

**Figure 14.** The $\chi^2$ probability of a dipole axis in the galaxy spin directions when 1% of the counter-clockwise galaxies are assigned with clockwise spin direction.

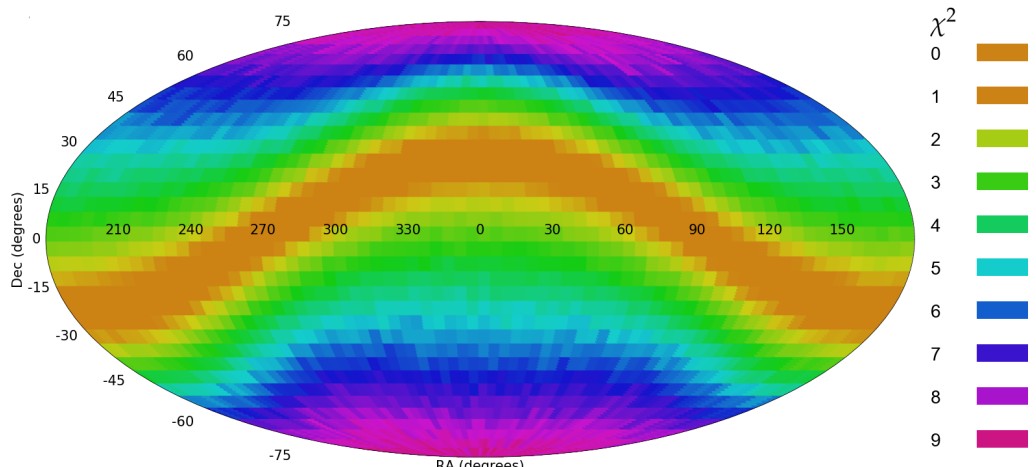

**Figure 15.** The $\chi^2$ probability of a dipole axis in spin directions when 2% of the counterclockwise galaxies are assigned with clockwise spin direction.

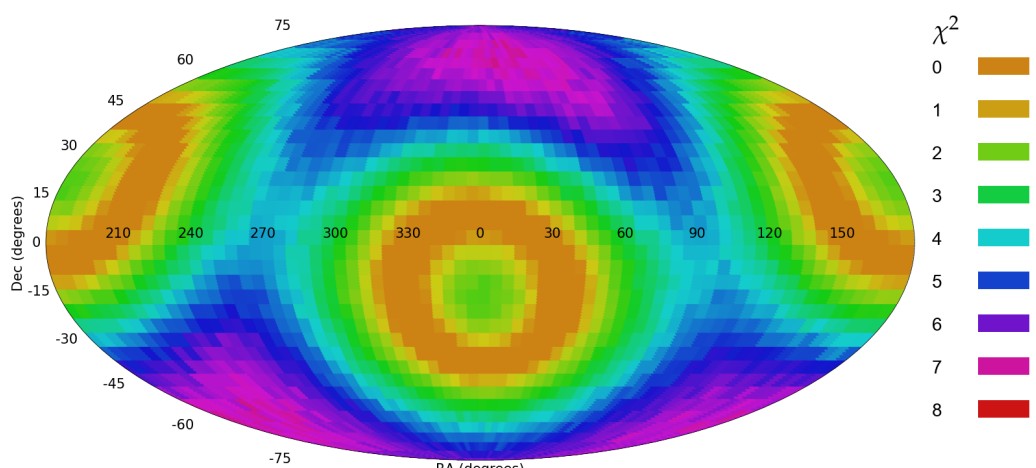

**Figure 16.** The $\chi^2$ probability of a quadrupole axis in spin directions when 2% of the counterclockwise galaxies are assigned with clockwise spin direction.

## 6. Non-Random Distribution of the Asymmetry

The method for profiling cosine dependence in spin direction of galaxies discussed in Section 2 can identify the location and statistical strength of a dipole axis if such exists. However, non-random distribution of the spin directions of spiral galaxies that does not necessarily form a dipole axis can also exhibit itself as a statistically significant dipole axis. To test the identification of a dipole, all galaxies in the dataset were assigned random spin directions. The only exception was galaxies in the sky region of ($120° < \alpha < 140°$, $0° < \delta < 20°$), in which all 2558 in that sky region galaxies were assigned with clockwise spin direction. Figures 17 and 18 show the probability of dipole and quadrupole axes in that dataset, respectively. As the figures show, the non-random distribution of these 2558 galaxies among the rest of the galaxies that were assigned random spin directions show strong statistical signal. The most likely dipole axis was identified with statistical strength of $15.049\sigma$, while quadrupole fitness showed a slightly weaker statistical significance of $14.029\sigma$. The strong signal is expected due to the very low probability of a large number of galaxies to have the same spin direction. But despite the fact that the rest of the sky showed no dipole axis, the small region of non-random spin directions was sufficient to lead to dipole and quadrupole axes with high statistical significance.

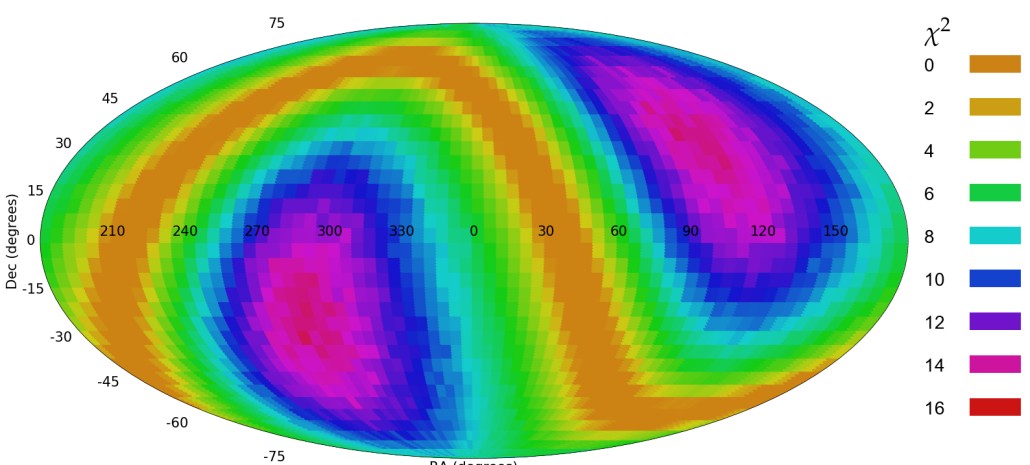

**Figure 17.** The $\chi^2$ probability of a dipole axis such that the galaxies are assigned with random spin directions, except for a certain sky region.

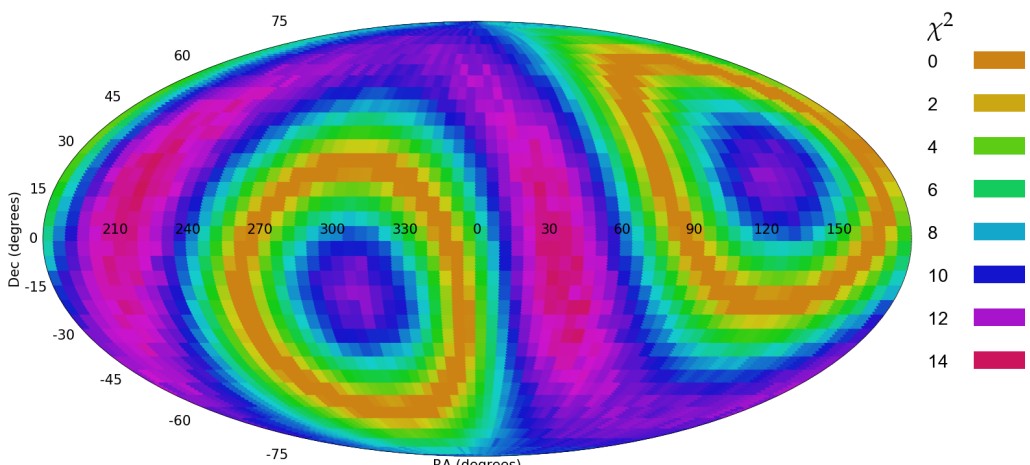

**Figure 18.** The $\chi^2$ probability of a quadrupole axis such that the galaxies are assigned with random spin directions, except for a certain sky region.

Figures 19 and 20 show a similar experiment, such that in addition to the galaxies in ($120° < \alpha < 140°$, $0° < \delta < 20°$), the 2336 galaxies in the sky region ($180° < \alpha < 200°$, $0° < \delta < 20°$) were also assigned with clockwise spin directions. That leads to a distribution of the galaxy spin directions that is not random, but also does not form a perfect dipole alignment. Figures 19 and 20 show the statistical significance of a dipole and quadrupole alignment in that dataset. The dipole axis had a statistical signal of $21.51\sigma$, and the quadrupole axis had $22.14\sigma$. That shows that non-random distribution of the galaxy spin directions can exhibit itself in the form of a dipole or quadrupole axes, also in case that the distribution of the spin directions of most galaxies in the dataset do not fit cosine dependence.

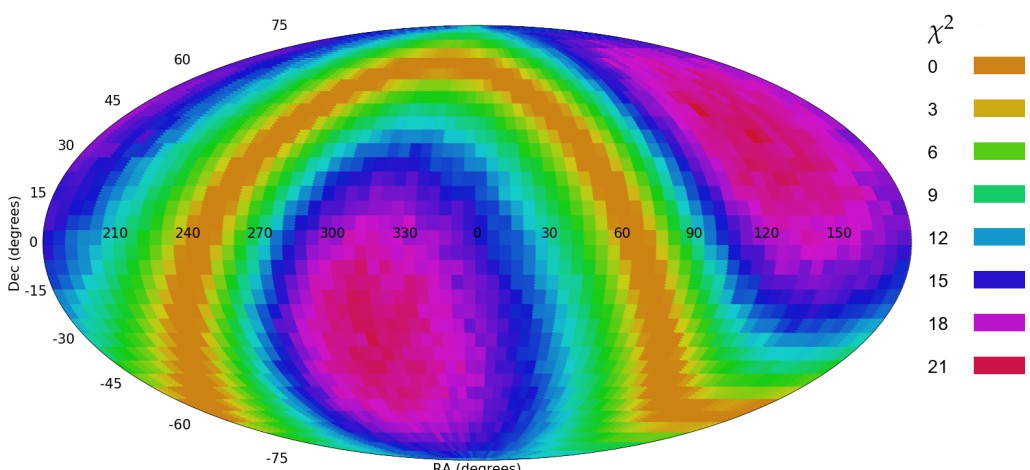

**Figure 19.** The $\chi^2$ probability of a dipole axis such that the galaxies are assigned with random spin directions, except for two sky regions that do not necessarily form a dipole.

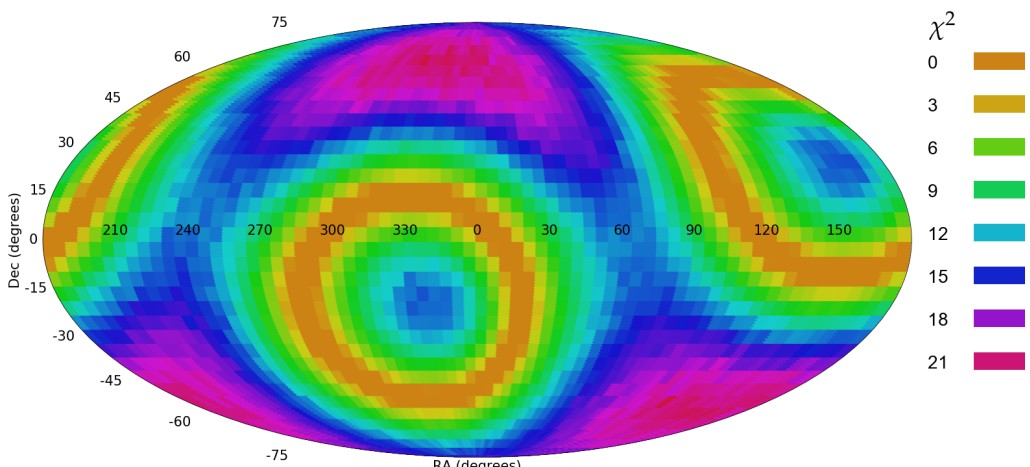

**Figure 20.** The $\chi^2$ probability of a quadrupole axis such that the galaxies are assigned with random spin directions, except for two different sky regions.

## 7. Conclusions

If the spin directions of spiral galaxies are aligned in the form of a dipole axis, the spin directions are expected to exhibit cosine dependence with the direction of observation compared to the location of the most likely axis. Here, a method that can identify the location of the most likely dipole axis by analyzing the spin directions of spiral galaxies is discussed. The method is tested in the light of potential anomalies in the data that can lead to a false detection of such dipole, or change its statistical signal.

The experiments show that duplicate objects in the dataset can increase the statistical signal of the detection of a dipole alignment of the distribution of spin directions of spiral galaxies. However, if the galaxy spin directions are distributed randomly, the duplicate objects need to make the dataset far larger than the original dataset. In the case of the dataset of SDSS galaxies tested here, each galaxy needed to have five duplicate objects to reach statistical significance when the galaxies were assigned with random spin directions. However, since duplicate object can artificially change the statistical signal, the analysis of cosine dependence or any other non-random distribution of spin directions of galaxies should be done in a dataset in which duplicate objects are removed.

Inaccuracy of the galaxy annotations does not have a substantial impact on the analysis of a dipole axis. Even if 25% of the galaxies are annotated randomly the statistical signal of the asymmetric distribution can still be identified. In any case, inaccurate annotations of the galaxy spin directions reduces the statistical signal of a possible dipole axis rather than increasing it.

The relatively weak impact of inaccuracy of the annotation of the spin directions is true only when the inaccurate annotations are distributed randomly between clockwise and counterclockwise galaxies. When the inaccurate annotations of the galaxies are not random, but have a systematic bias toward a certain spin direction, even a relatively small bias can lead to a statistically significant dipole axis detected in the dataset. For instance, even a small consistent bias of 1% of the annotations leads to a statistically significant dipole axis. That artificial axis peaks at the celestial pole, which is expected since SDSS contains galaxies mostly from the Northern hemisphere. In general, dipole axes that peak at the celestial pole should be examined with caution, as many catalogs are created by using ground-based instruments that cannot cover more than one hemisphere. When combining data from different hemispheres into a single catalog, a dipole of asymmetry that is aligned with the celestial pole might indicate on differences between the instrument that collected the data from the Northern hemisphere and the instrument that collected the data from the Southern hemisphere. Such difference is expected to exhibit itself in the form of a dipole axis that peaks at the celestial pole, but does not necessarily indicate on an astronomical or cosmological phenomenon.

The analysis used here is also sensitive to non-random distribution of galaxy spin directions even if the spin directions are not aligned in the form of a dipole axis. For instance, a single spot in the sky with very strong asymmetry in galaxy spin directions can lead to the identification of a dipole axis in that spot with very strong statistical signal, even if the spin directions of all other galaxies in the dataset are distributed randomly. While a cosmological dipole axis of asymmetry in galaxy spin directions is expected to exhibit itself in the form of cosine dependence, other anomalies in the distribution of spin directions of spiral galaxies can also be identified in the form of a statistically significant cosine dependence. Therefore, full identification of a possible dipole or quadrupole axes will require the analysis of a very high number of galaxies covering a large part of the sky to fully profile the nature of a possible asymmetry in the spin directions of spiral galaxies.

The dataset that was used in this study is a dataset of SDSS galaxies designed for experiments related to photometry of galaxies [21]. Here the dataset is tested for the identification of non-random distribution of galaxy spin directions, and for that purpose photometric objects that are part of the same galaxies were removed. The statistical signal after removing the duplicate objects is $2.56\sigma$ for a dipole axis, and $3.0\sigma$ for a quadrupole axis. The statistical signal does not meet the $5\sigma$ discovery threshold, but it is still considerable, and comparable to the statistical signal of other provocative observations of primary scientific interest such as the CMB cold spot [24]. The analysis also agrees with previous experiments using automatic annotation of galaxy images [2,6] showing non-random distribution of galaxies with opposite spin directions.

The most likely dipole axis was identified at ($\alpha = 165°$, $\delta = 40°$), with $1\sigma$ error range of ($90°$, $240°$) for the RA and ($-35°$, $90°$) for the declination. The most likely dipole axis reported with a dataset of spectorscopic objects reported in [2] is ($\alpha = 132°, \delta = 32°$), close to the dipole axis shown here, and within $1\sigma$ error. The axis reported by Longo [1] at ($\alpha = 217°, \delta = 32°$), which somewhat more distant, but also within the $1\sigma$ error from the dipole axis reported here. The dipole axis shown with Galaxy Zoo data at ($161°$, $11°$) is also within close distance to the dipole axis shown here, although it should be noted that the asymmetry between clockwise and counterclockwise galaxies in Galaxy Zoo data was determined to be driven by perceptional bias of the volunteers who annotated the data, and not statistically significant when the perceptual bias was corrected [8].

Given the accumulating evidence for cosmological-scale anisotropy [25–31], the observations reinforce to continue the investigation for better understanding whether galaxy spin directions indeed form pattern of non-random distribution. Non-random distribution of spiral galaxies exhibiting a dipole or quadrupole axes is naturally difficult to explain with the current "mainstream" cosmological theories. However, evidence for cosmological-scale anisotropy and possible existence of cosmological-scale axes have been observed to certain extent in the cosmic microwave background [32–34], also leading to theories that shift from the standard model [35–40].

An observation of a cosmological-scale dipole axis can be related to non-standard theories such as ellipsoidal universe [41–43], or rotating universe [44–48], as the possible spin in the large-scale structure might exhibit itself in the large-scale correlation in the spin direction of the galaxies, and form a cosmological-scale axis. The existence of a cosmological-scale axis in the spin directions of galaxies can also be aligned with theories such as holographic big bang model [49,50]. Since a black hole is expected to spin [51], a cosmological-scale axis in the spin directions of galaxies can agree with the expected spin of the host black hole.

Clearly, more research will be needed to test the possible non-random distribution of galaxies with opposite spin directions, and profile possible patterns that it exhibits if such non-random distribution indeed exists. While the analysis discussed here is based on SDSS data, these observations are aligned with smaller datasets from Pan-STARRS [6] and Hubble Space Telescope [52]. Future analysis will include larger datasets such as the Dark Energy Survey and the Vera Rubin Observatory, providing far larger and deeper datasets. The multiple observations using different methods and different instruments reporting on

such anomalies [1–7,9] reinforce the studying and profiling of the observations, as well as examining non-astronomical reasons that can lead to such observations.

**Funding:** This research received no external funding.

**Institutional Review Board Statement:** Not applicable.

**Informed Consent Statement:** Not applicable

**Data Availability Statement:** Data available in a publicly accessible repository that does not issue DOIs. This data can be found here: http://people.cs.ksu.edu/~lshamir/data/assymdup.

**Acknowledgments:** I would like to thank the two knowledgeable anonymous reviewers for the insightful comments that helped to improve the manuscript. This study was supported in part by NSF grants AST-1903823 and IIS-1546079. SDSS-IV is managed by the Astrophysical Research Consortium for the Participating Institutions of the SDSS Collaboration including the Brazilian Participation Group, the Carnegie Institution for Science, Carnegie Mellon University, the Chilean Participation Group, the French Participation Group, Harvard-Smithsonian Center for Astrophysics, Instituto de Astrofisica de Canarias, the Johns Hopkins University, Kavli Institute for the Physics and Mathematics of the Universe (IPMU)/University of Tokyo, the Korean Participation Group, Lawrence Berkeley National Laboratory, Leibniz Institut fur Astrophysik Potsdam (AIP), Max-Planck-Institut fur Astronomie (MPIA Heidelberg), Max-Planck-Institut fur Astrophysik (MPA Garching), Max-Planck-Institut fur Extraterrestrische Physik (MPE), National Astronomical Observatories of China, New Mexico State University, New York University, University of Notre Dame, Observatario Nacional/MCTI, the Ohio State University, Pennsylvania State University, Shanghai Astronomical Observatory, United Kingdom Participation Group, Universidad Nacional Autonoma de Mexico, University of Arizona, University of Colorado Boulder, University of Oxford, University of Portsmouth, University of Utah, University of Virginia, University of Washington, University of Wisconsin, Vanderbilt University, and Yale University.

**Conflicts of Interest:** The author declares no conflict of interest.

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
