# Peer review of "Analysis of the Alignment of Non-Random Patterns of Spin Directions in Populations of Spiral Galaxies"

_2571-712X, doi:10.3390/particles4010002_

Round 1
Reviewer 1 Report
Dear editor, Dear author,
The manuscript on galaxy spin directions by L. Shamir is an interesting study looking at how various factors can create or weaken a signal of galaxy spins being preferentially aligned, which is not expected for the Universe as a whole due to the assumption of isotropy. The manuscript has some potentially valuable new insights regarding how various observational issues could affect analyses of this sort, which have been attempted in the past with application to real data. The present manuscript instead uses idealised situations to gain an insight into how controlled adjustments to the data would affect the analysis. These controlled experiments are useful, and overall I recommend the manuscript be accepted in close to its current form. However, I have some recommendations to improve the manuscript before that time.
My main recommendation is that the authors mention in the introduction about the broader context of their work, in particular with relation to the length scale over which correlations between galaxy spin directions might be expected. The possibility of correlations between galaxy spins and large scale structure in the standard LCDM cosmology should be discussed, for instance using this work:
https://doi.org/10.3847/1538-4357/aae20f
Whether galaxies are too far apart to influence each other gravitationally depends on the assumed gravity law, with stronger gravity at long-range presumably increasing this length and leading to stronger correlations. I therefore recommend the authors discuss this work:
https://dx.doi.org/10.1093/mnras/staa2348
It will help to show that there is evidence that gravitational interactions may be of a longer range nature than expected in standard cosmology, which would help to explain the Hubble tension by means of our location within a particularly wide and deep supervoid of a sort not expected in LCDM. However, such a void is observed and could well arise in a Milgromian cosmology, as discussed above. The main reason this is possible is that the expansion rate history, cosmic microwave background anisotropies, and primordial light element abundances would remain the same as in LCDM, but structure formation would be enhanced due to the modified gravity law. Enhanced structure formation compared to LCDM is also suggested by El Gordo, whose observed properties falsify LCDM at 6.16 sigma:
https://dx.doi.org/10.1093/mnras/staa3441
Thus, correlations between galaxy spin directions may hold important clues to some fundamental issues like the large scale behaviour of gravity.
Minor comments:
Section 2: mathematical variables like d should be italicised, or put in math mode $d$. I also recommend to use N_{cw} and N_{acw} to denote the number of galaxies spinning clockwise and anti-clockwise, respectively.
Figures 3, 6, 12: some variable (A?) should be defined in the text and this should be used as the y-label. The captions can then refer to the equation defining A (for asymmetry), so readers can jump to that to understand what is plotted on the y-axis. This will avoid having to define this every time.
Figures 4, 5, 9, 11, 13 – 19: The label on the colour bar should use \chi, as \sigma is typically used for uncertainty. What the authors really mean is the statistical significance expressed as the equivalent number of standard deviations for a one-dimensional Gaussian, which is better expressed as \chi. This should also be clarified somewhere in the text, or in the caption the first time this is used.
Page 4: the authors should use e.g. P = 0.0017, and typically do not need to enclose this in brackets. Also, the authors should use = when meaning equality to several significant figures, and \approx if only the first significant figure is known. \sim (or ~) should only be used if the authors mean that only the order of magnitude of the result is known, but it is clear that the authors almost never mean this – yet they liberally use ~. This is not good.
Section 7: Use ‘Conclusions’
In summary, the manuscript would benefit from some minor adjustments before publication, especially to discuss non-standard theories which may enhance correlations between galaxy spin directions beyond that expected in the standard LCDM model.
Yours sincerely,
Referee
Author Response
The manuscript on galaxy spin directions by L. Shamir is an interesting study looking at how various factors can create or weaken a signal of galaxy spins being preferentially aligned, which is not expected for the Universe as a whole due to the assumption of isotropy. The manuscript has some potentially valuable new insights regarding how various observational issues could affect analyses of this sort, which have been attempted in the past with application to real data. The present manuscript instead uses idealised situations to gain an insight into how controlled adjustments to the data would affect the analysis. These controlled experiments are useful, and overall I recommend the manuscript be accepted in close to its current form. However, I have some recommendations to improve the manuscript before that time.
--Author response: I would like to thank you for the time you invested in reviewing the paper, and for the insightful comments. Many changes have been made based on the comments. The reply to each comment with a description of the consequent changes made to the manuscript are below, next to each comment. For convenience, changes in the manuscript were made in bold font.
My main recommendation is that the authors mention in the introduction about the broader context of their work, in particular with relation to the length scale over which correlations between galaxy spin directions might be expected. The possibility of correlations between galaxy spins and large scale structure in the standard LCDM cosmology should be discussed, for instance using this work:
https://doi.org/10.3847/1538-4357/aae20f
--Author response: Thank you for the comment! That is actually an important recent paper that I was not aware of. I added to the introduction a discussion about that paper and related work, including several relevant papers that were cited in that paper.
Whether galaxies are too far apart to influence each other gravitationally depends on the assumed gravity law, with stronger gravity at long-range presumably increasing this length and leading to stronger correlations. I therefore recommend the authors discuss this work:
https://dx.doi.org/10.1093/mnras/staa2348
--Author response: Thank you for the comment. The papers that I cited assume Newtonian gravity, which is indeed an assumption, and other options should be also considered. It is indeed possible that if the Milgromian model is correct, it can also explain the links between the spin directions of galaxies too far apart to have *Newtonian* gravitational links. That is an important point, and a reference to the paper and a discussion about it has been added to the Introduction section (in bold font).
It will help to show that there is evidence that gravitational interactions may be of a longer range nature than expected in standard cosmology, which would help to explain the Hubble tension by means of our location within a particularly wide and deep supervoid of a sort not expected in LCDM. However, such a void is observed and could well arise in a Milgromian cosmology, as discussed above. The main reason this is possible is that the expansion rate history, cosmic microwave background anisotropies, and primordial light element abundances would remain the same as in LCDM, but structure formation would be enhanced due to the modified gravity law. Enhanced structure formation compared to LCDM is also suggested by El Gordo, whose observed properties falsify LCDM at 6.16 sigma:
https://dx.doi.org/10.1093/mnras/staa3441
--Author response: That’s another good point that I did not think of. Thank you for making this comment. I added a detailed paragraph to the Conclusions section with exactly that discussion, and added the reference. I believe that makes the paper stronger, and I thank you for making that comment.
Thus, correlations between galaxy spin directions may hold important clues to some fundamental issues like the large scale behaviour of gravity.
--Author response. Yes. That has been added to the Conclusions section as described above.
Minor comments:
Section 2: mathematical variables like d should be italicised, or put in math mode $d$. I also recommend to use N_{cw} and N_{acw} to denote the number of galaxies spinning clockwise and anti-clockwise, respectively.
--Author response: Thank you for the comment. That has been corrected as suggested.
Figures 3, 6, 12: some variable (A?) should be defined in the text and this should be used as the y-label. The captions can then refer to the equation defining A (for asymmetry), so readers can jump to that to understand what is plotted on the y-axis. This will avoid having to define this every time.
--Author response: Thank you the suggestions. The figures have been corrected as suggested, and the definition of the variable A is in the text as well as the caption of the first figure (Figure 3), as suggested.
Figures 4, 5, 9, 11, 13 – 19: The label on the colour bar should use \chi, as \sigma is typically used for uncertainty. What the authors really mean is the statistical significance expressed as the equivalent number of standard deviations for a one-dimensional Gaussian, which is better expressed as \chi. This should also be clarified somewhere in the text, or in the caption the first time this is used.
--Author response: The labels in all of these figures have been modified as suggested.
Page 4: the authors should use e.g. P = 0.0017, and typically do not need to enclose this in brackets. Also, the authors should use = when meaning equality to several significant figures, and \approx if only the first significant figure is known. \sim (or ~) should only be used if the authors mean that only the order of magnitude of the result is known, but it is clear that the authors almost never mean this – yet they liberally use ~. This is not good.
--Author response: I agree that the use of P and ~ was not perfect. That has been corrected in the revised version.
Section 7: Use ‘Conclusions’
--Author response: Corrected. Thank you.
In summary, the manuscript would benefit from some minor adjustments before publication, especially to discuss non-standard theories which may enhance correlations between galaxy spin directions beyond that expected in the standard LCDM model.
--Author response: In the conclusion section I added some discussion about non-standard theories and how a cosmological-scale axis in spin direction can fit in them. The theory of a rotating universe and the theory of holographic big bang are two theories that I discuss, with an explanation of how such axis can agree with these theories. I offer no opinion of whether these theories are true or not, but just discuss them in the light of the findings as theories that were proposed in the existing literature.
Yours sincerely,
Referee
Thank you for the helpful comments!
Reviewer 2 Report
REVIEWERS REPORT – Analysis of the Alignment of Non-random Patterns of Spin Directions in Populations of Spiral Galaxies by Lior Shamir
Shamir's article is potentially of considerable interest to cosmology. However, I do not believe it should be published in its present form.
My main objection has to do with his plots in Figs. 3, 6, 8, 10. Here he plots spin asymmetry vs. RA. The points for opposite hemispheres are connected by orange or blue lines. It is not at all clear what these lines represent. They seem to mostly go right through the points. The symbols for the points on the graphs for the two hemispheres are not distinguishable. It is clear at a glance that the points are strongly correlated and the error bars are misleading at best. This makes the cosine dependence look a lot more significant than it really is. The overall effect is confusing and misleading to a reader who does not delve into the details.
A much more appropriate way to plot these data is a simple asymmetry binned in RA segments as averaged over all declinations covered by SDSS with appropriate 1/sqrt(N) errors along with a conventional chisq fit to a cosine dependence.
His two-dimensional chisq plots as in Fig. 5 are also confusing in that the best fit axis of (165°, 40°) will have an equally good fit 180° away at (345°, -40°). Perhaps this can be alleviated by just plotting one hemisphere centered at RA=165°. These can also be based on a simple asymmetry binned in RA. I would also like to see a little more discussion of a possible dependence of the asymmetry on galaxy brightness since, if there is a real asymmetry it would tend to be washed out as Ganalyzer makes more spin misclassifications for fainter galaxies. [There's also the possibility that a selection bias might become more apparent for fainter galaxies.]
To a great extent Shamir is belaboring the obvious in this article. – Yes, not surprisingly adding extra duplicated points reduces the statistical uncertainty. Yes, when random spin assignments are made they do tend to wash out any real asymmetry. Yes, if biased spin assignments are assigned to only one angular region it will cause an increase (or decrease) in the dipole signal.
I believe the changes I suggest will result in a much more compelling case for a possible real asymmetry. Shamir does emphasize the need for more data in the southern galactic hemisphere, and I wholeheartedly agree. If the apparent spin asymmetry in the SDSS data is mirrored in the other hemisphere, it would go a long way in demonstrating that the asymmetry is not due to a bias as well as improving its statistical significance. I don't believe that the Dark Energy Survey images have been made publicly available yet. Hopefully they will be soon.
Author Response
Shamir's article is potentially of considerable interest to cosmology. However, I do not believe it should be published in its present form.
--Author response: Thank you for reviewing the paper and for the insightful comments. Substantial changes have been made to the manuscript based on these comments (in the relatively short time I was given to revise it). The replies to the comments and description of the changes made to the manuscript are listed below. Changes to the manuscript were made in bold font.
My main objection has to do with his plots in Figs. 3, 6, 8, 10. Here he plots spin asymmetry vs. RA. The points for opposite hemispheres are connected by orange or blue lines. It is not at all clear what these lines represent. They seem to mostly go right through the points. The symbols for the points on the graphs for the two hemispheres are not distinguishable. It is clear at a glance that the points are strongly correlated and the error bars are misleading at best. This makes the cosine dependence look a lot more significant than it really is. The overall effect is confusing and misleading to a reader who does not delve into the details.
A much more appropriate way to plot these data is a simple asymmetry binned in RA segments as averaged over all declinations covered by SDSS with appropriate 1/sqrt(N) errors along with a conventional chisq fit to a cosine dependence.
--Author response: Thank you for the comment. I agree that the two curves could be somewhat ambiguous, and definitely contain redundant information. The redundancy is intentional, but still could be confusing. I absolutely agree that the visualization might seem like cosine dependence despite the fact that that specific simple analysis does not necessarily show that. It does not aim at showing that, but as you commented above, a reader who does not look carefully at all the details might be misled to believe that the graphs provide evidence of cosine dependence. All of the figures mentioned in the comment have been replaced as proposed. The new figures actually also show certain evidence of cosine dependence, but these figures show separate sky sections, and not any kind of sliding windows. These figures are made directly from the data (which is made public and the URL to access the data is in the manuscript), and show separate bins such that no galaxy can exist in more than one bin. I agree that it is a more reliable representation of the data. It shows some certain evidence of cosine dependence, but that is due to the data and not the method of visualization.
His two-dimensional chisq plots as in Fig. 5 are also confusing in that the best fit axis of (165°, 40°) will have an equally good fit 180° away at (345°, -40°). Perhaps this can be alleviated by just plotting one hemisphere centered at RA=165°. These can also be based on a simple asymmetry binned in RA. I would also like to see a little more discussion of a possible dependence of the asymmetry on galaxy brightness since, if there is a real asymmetry it would tend to be washed out as Ganalyzer makes more spin misclassifications for fainter galaxies. [There's also the possibility that a selection bias might become more apparent for fainter galaxies.]
--Author response: A new figure (Figure 5) has been added. That figure is the Mollweide projection centered at (165o,40o), and includes just one hemisphere. It might not be as intuitive as the other figure, so I left both figures there. I can remove the full sky image if really needed, but as you can see, the full sky projection is more intuitive, and also used commonly. Half hemisphere is used less often, and as you can see it still shows evidence of the two sides of the sky being analyzed in the same fashion. The full sky projection is also made without an assumption, while centering around (165o,40o) is based on the initial identification that the axis peaks in that point. The full sky is made with no preliminary information. But now both images are there. The existence of the half-hemisphere image can attract the attention of the reader to understand the details of the analysis, and I also added an explanation that the full sky image is made of two hemispheres that are analyzed in the same way and the same data.
The simple binned asymmetry in different RA ranges is shown in the new Figure 3.
Analysis of fainter and brighter galaxies was also made, and new figures (Figures 9 and 10 in the revised manuscript) and discussion have been added to the manuscript. That was done by separating the galaxies to two orthogonal datasets g<18 and g>18. It is interesting that when separating the dataset to brighter and fainter galaxies, and no galaxy exists in both datasets, both sets show statistically significant asymmetry, with a dipole axis at roughly the same part of the sky.
In any case, spin misclassification is not expected to lead to signal, as shown through the experiments in Section 5. But I also added the theoretical explanation with a few equations to show that misclassification of galaxies is not expected to lead to increased signal, but in fact to weaker signal. That has been added also to Section 5. For convenience, the new paragraphs are written in bold font.
To a great extent Shamir is belaboring the obvious in this article. – Yes, not surprisingly adding extra duplicated points reduces the statistical uncertainty. Yes, when random spin assignments are made they do tend to wash out any real asymmetry. Yes, if biased spin assignments are assigned to only one angular region it will cause an increase (or decrease) in the dipole signal.
--Author response: I completely agree with the comment. The reason for discussing all of these aspects, which I agree are obvious, is because these are all actual responses I received in the past, either at conferences or through direct communication. I agree that they are all obvious, and it even somewhat frustrates me to discuss such obvious points. But these are all arguments brought up by our colleagues. It is even the main topic of the paper to address all of these arguments and analyze their impact quantitatively. I did remove one figure (Figure 11 in the original version) and shortened the discussion.
I believe the changes I suggest will result in a much more compelling case for a possible real asymmetry. Shamir does emphasize the need for more data in the southern galactic hemisphere, and I wholeheartedly agree. If the apparent spin asymmetry in the SDSS data is mirrored in the other hemisphere, it would go a long way in demonstrating that the asymmetry is not due to a bias as well as improving its statistical significance. I don't believe that the Dark Energy Survey images have been made publicly available yet. Hopefully they will be soon.
--Author reply: DES is not as mature as SDSS, and it will probably take some time until the data become available in a similar form. Vera Rubin Observatory will provide an even larger dataset. But so far we have analysis from SDSS, Pan-STARRS, and HST, all showing similar profiles. I added a comment and relevant references to the conclusion section, summarizing the telescopes tested so far, and relevant telescopes that will be tested in the future.
Round 2
Reviewer 2 Report
REVIEWERS REPORT II – Analysis of the Alignment of Non-random Patterns of Spin Directions in Populations of Spiral Galaxies by Lior Shamir
Shamir's rewrite is a big improvement. I believe it should be published though it needs some more work. My main questions have to do with Fig. 3 but let me go through the text in order of appearance.
–A sentence or two might be added in the abstract that conclusions agree with previous work by Shamir and by others.
–In the text around Line 44 it should be made clear that the (one-headed vector) spin is not the same as an (two-headed) alignment for which the spins show no preference between parallel and antiparallel to the direction of the alignment.
–Around Line 122, "That shows that SDSS galaxies are not evenly distributed in the sky, and the population varies significantly in different RA and declination ranges." should be replaced with something like "This is because most of the SDSS galaxies are in the hemisphere toward right ascension 180°." He should also make it clear that a real asymmetry might appear as a "bias" because of this.
–In Fig. 3 I don't understand what the fat red bars represent and they do not seem to be explained in the text. ??
–In Fig. 3 I don't understand why Shamir can't show the best fit cos(RA-165°) curve. The cosine dependence is a main thrust of his article yet he never shows it explicitly in his figures.
–I don't understand why he doesn't show the whole plot in Fig. 5.
–The discussion around Line 133 is confusing. The data are the same, except for sign, whether you look from (165°, 40°) or from the opposite direction (345°, -40°).
–The discussion of gravitational alignments around Line 312 is very confusing. Gravitational interactions cannot cause a parity-violating spin asymmetry while tidal effects can certainly cause alignments along gravitational gradients. I suggest this discussion be removed or modified substantially, as it detracts from Shamir's main thesis of a parity-violating asymmetry and a possible breakdown of cosmic isotropy. –Around Line 127 Shamir compares his best fit axis (165°, 40°) with that from his previous work (132°, 32°). He should also compare it to Longo's best fit axis (217°± 35°, 32°) and Galaxy Zoo's (161°,11°). I strongly recommend that this discussion be moved to the Conclusions section where Shamir can make a comprehensive comparison of all the results including their statistical significance. [In fact, they agree quite well as they should since all are using pretty much the same data.] This presents a great opportunity for him to summarize the evidence for a real asymmetry and it will go a long way to give readers confidence in the overall results since the methods and possible biases of the three studies are very different.
Author Response
I would like to thank the anonymous referee once again for the insightful comments, and also for the timely response. I agree with all comments and changed the paper accordingly, but I am mostly grateful for the last comment, which I believe is indeed important, and makes the paper stronger. All comments have been addressed, and the replies to the specific comments and description of the changes made to the manuscript are listed below.
Sincerely,
Lior
REVIEWERS REPORT II – Analysis of the Alignment of Non-random Patterns of Spin Directions in Populations of Spiral Galaxies by Lior Shamir
Shamir's rewrite is a big improvement. I believe it should be published though it needs some more work. My main questions have to do with Fig. 3 but let me go through the text in order of appearance.
–A sentence or two might be added in the abstract that conclusions agree with previous work by Shamir and by others.
--Author response: I added a short sentence in the end of the abstract as suggested. I cannot add references to the abstract, but I added a broad statement about the agreement with previous work and different telescopes, and the full information is in the Conclusions section of the paper.
–In the text around Line 44 it should be made clear that the (one-headed vector) spin is not the same as an (two-headed) alignment for which the spins show no preference between parallel and antiparallel to the direction of the alignment.
--Author response: Thank you. A note has been added to clarify that point.
–Around Line 122, "That shows that SDSS galaxies are not evenly distributed in the sky, and the population varies significantly in different RA and declination ranges." should be replaced with something like "This is because most of the SDSS galaxies are in the hemisphere toward right ascension 180°." He should also make it clear that a real asymmetry might appear as a "bias" because of this.
--Author response: That’s a good point. The text has been changed as proposed (the change is in bold font).
–In Fig. 3 I don't understand what the fat red bars represent and they do not seem to be explained in the text. ??
--Author response: That figure was requested by the other reviewer. It shows the asymmetry between the number of clockwise and counterclockwise galaxies in each RA range, when the declination of the galaxies is ignored. The declination range in SDSS is not very broad, so the simple visualization can be useful. I added more description about Figure 3 (around line 114, in bold font), and it also includes formal definition of the asymmetry. I also added some more information in the caption.
–In Fig. 3 I don't understand why Shamir can't show the best fit cos(RA-165°) curve. The cosine dependence is a main thrust of his article yet he never shows it explicitly in his figures.
--Author response: The cosine line has been added to the figure as suggested.
–I don't understand why he doesn't show the whole plot in Fig. 5.
--Author response: Figure 5 was changed. It now includes the entire sky.
–The discussion around Line 133 is confusing. The data are the same, except for sign, whether you look from (165°, 40°) or from the opposite direction (345°, -40°).
--Author response: I agree that the discussion is confusing, and does not contribute much to the manuscript. I removed that sentence from the revised version.
–The discussion of gravitational alignments around Line 312 is very confusing. Gravitational interactions cannot cause a parity-violating spin asymmetry while tidal effects can certainly cause alignments along gravitational gradients. I suggest this discussion be removed or modified substantially, as it detracts from Shamir's main thesis of a parity-violating asymmetry and a possible breakdown of cosmic isotropy. –Around Line 127 Shamir compares his best fit axis (165°, 40°) with that from his previous work (132°, 32°). He should also compare it to Longo's best fit axis (217°± 35°, 32°) and Galaxy Zoo's (161°,11°). I strongly recommend that this discussion be moved to the Conclusions section where Shamir can make a comprehensive comparison of all the results including their statistical significance. [In fact, they agree quite well as they should since all are using pretty much the same data.] This presents a great opportunity for him to summarize the evidence for a real asymmetry and it will go a long way to give readers confidence in the overall results since the methods and possible biases of the three studies are very different.
--Author response: Thank you for the comment! I agree that it is a good idea to make the comparison, and a paragraph about it has been added to the conclusion section as suggested. I had to make a note that the Galaxy Zoo asymmetry was argued by the authors to be not statistically significant because that’s what the authors claim in their paper. But the dipole, whether significant or not, is indeed very close to the dipole here. I was not aware of that (did not think to compare to GZ), and I thank you for the comment.
I also removed the paragraph about the gravitational alignment (around line 312 in the previous version of the manuscript) as suggested. I agree that it somewhat shifts from the main point.